# Mechanism of Tumor Budding in Patient-Derived Metachronous Oral Primary Squamous Cell Carcinoma Cell Lines

**DOI:** 10.3390/ijms26073347

**Published:** 2025-04-03

**Authors:** Takayuki Omae, Yuji Omori, Yuna Makihara, Koji Yamanegi, Soutaro Hanawa, Kyohei Yoshikawa, Kazuma Noguchi, Hiromitsu Kishimoto

**Affiliations:** 1Department of Oral and Maxillofacial Surgery, School of Medicine, Hyogo Medical University, 1-1 Mukogawa-cho, Nishinomiya 663-8501, Japan; ta-oomae@hyo-med.ac.jp (T.O.); yu-oomori@hyo-med.ac.jp (Y.O.); yu-makihara@hyo-med.ac.jp (Y.M.); so-hanawa@hyo-med.ac.jp (S.H.); kyo331@hyo-med.ac.jp (K.Y.); kisihiro@hyo-med.ac.jp (H.K.); 2Department of Pathology, School of Medicine, Hyogo Medical University, Nishinomiya 663-8501, Japan; yamanegi@hyo-med.ac.jp

**Keywords:** tumor budding, metachronous oral squamous cell carcinoma, epithelial–mesenchymal transition, transforming growth factor-β

## Abstract

Tumor budding (TB) occurs at the deepest site of tumor invasion and is a significant prognostic indicator of cervical metastasis in oral squamous cell carcinoma (OSCC). The mechanism of TB, however, remains unclear. This study investigated the roles of the tumor microenvironment and partial epithelial–mesenchymal transition (p-EMT) in TB expression using molecular and cellular physiological analyses. We established oral metachronous carcinoma cell lines (gingival carcinoma: 020, tongue carcinoma with high TB expression: 020G) from two cancers with pathologically different TB in the same patient and subjected them to exome analysis to detect gene mutations related to carcinogenesis and malignancy. Differences in EMT expression induced by transforming growth factor-β (TGF-β) between 020 and 020G were analyzed by Western blotting and reverse transcription polymerase chain reaction, and TGF-β-induced changes in cell morphology, proliferation, migration, and invasive ability were also examined. TGF-β expression was observed in the deepest tumor invasion microenvironment. TGF-β also induced the expression of several p-EMT markers and increased the migration and invasive abilities of 020G compared with 020 cells. In conclusion, TGF-β in the deep-tumor microenvironment can induce p-EMT in tumor cells, expressed as TB.

## 1. Introduction

Oral squamous cell carcinoma (OSCC) is one of the most common malignant tumors, affecting approximately 370,000 patients annually worldwide by 2020 [1]. Most deaths result from local recurrence at the primary sites and from peripheral lymph node metastasis [2,3]. Metastasis to neck lymph nodes directly affects the prognosis of patients, regardless of the number, the size of metastatic lymph nodes, and the presence of extranodal extension [4], and 14–45% of patients who are clinically negative for neck metastasis (cN0) already have occult neck metastasis [5,6].

Tumor budding (TB) refers to the presence of detached single tumor cells or clusters of up to five cells scattered within the stroma at the invasive tumor front in many different solid cancers, including esophageal, laryngeal, and nasopharyngeal cancers [7,8,9,10]. TB and its association with disease progression in patients with various solid cancers was first described by Imai in the 1950s [11], and a few studies have evaluated TB in patients with cT1-T2N0 OSCC [12,13,14]. TB has also emerged as a promising prognostic biomarker across several different tumor types, including head and neck cancer, predicting disease progression and unfavorable survival [15]. Yamakawa et al. reported that TB was a significant risk factor for late cervical lymph node metastasis in patients with clinically early-stage squamous cell carcinoma (SCC) of the tongue [16]. These findings suggest that TB may be used as a predictor of cervical lymph node metastasis and thus prognosis in patients with oral cancer; in reality, however, TB cannot be detected by observing part of the tumor, such as in a biopsy, but only in a whole tumor sample, after radical surgery. Given the potential of TB as a marker of cervical lymph node metastasis in OSCC, and the need for cervical dissection followed by concurrent chemoradiotherapy (CCRT) for cervical control, it is necessary to identify biomarkers that can confirm the presence of TB to inform preoperative treatment decisions.

Biologically, TB occurs in interaction with the tumor microenvironment (TME) and is associated with epithelial–mesenchymal transition (EMT) [17]. OSCCs have been shown to overexpress EMT transcription factors (TFs), such as *SNAIL1/2*, *TWIST1*, and *ZEB1* [18,19]. EMT is thought to be a potential driver of invasiveness and metastasis in a variety of epithelial cancers [20]. Most studies about EMT have focused on cell lines and/or animal models, but Puram et al. [21,22] reported the presence of a subpopulation of epithelial cells expressing partial-EMT (p-EMT) as a hybrid state including both epithelial and mesenchymal characteristics simultaneously [23]. In addition, prospective validation of the p-EMT biomarkers podoplanin (PDPN), laminin subunit beta-3 (LAMB3), and laminin subunit gamma-2 (LAMC2) may aid decision-making regarding the need for neck dissection in the N0 neck and/or for adjuvant therapy [24]. Lee et al. [25] investigated EMT marker expression in lymph nodes with and without extranodal extension (ENE) and found that high expression of EMT markers in the ENE region was a more accurate predictor of poor prognosis and systemic metastasis than the presence of ENE alone.

Thus, it has recently become clear that TB, which appears at the deepest part of the infiltrate, may be a new prognostic factor in OSCC. Puram et al. [21,22] reported that p-EMT is not a classical EMT program, but rather a plastic metastable state with both mesenchymal properties while the adhesive epithelial markers are preserved. They also reported that the p-EMT program localizes to the tip of the primary tumor, allowing for population migration of tumor cells and proximity to cancer-associated fibroblasts within the TME, suggesting that it is closely associated with TB that appears at the deepest sites of tumor invasion. Based on these reports, we hypothesized that the occurrence of the phenomenon of TB, which appears in the deepest part of the tumor, may be expressed as a phenotype of p-EMT. Understanding the relationship between the pathogenesis of TB and p-EMT may help in the development of new therapeutic strategies for OSCC and may be used as a prognostic factor.

In this study, we successfully established cultured cell lines from metachronous oral cancers with different TB expression features in the same patient. We also analyzed the genetics of metachronous oral carcinomas and the factors determining p-EMT expression in relation to TB and the grade of cancer when the cancer microenvironment is reproduced in vitro for each cultured cell line.

## 2. Results

### 2.1. Case Presentation

A 66-year-old Japanese man presented to our department with a chief complaint of spontaneous pain and paralysis in the left mandibular gingiva. Intraoral findings revealed an ulcerative tumor with granulomatous tissue measuring 40 × 30 mm extending posteriorly from the left lower second premolar. Three significantly enlarged lymph nodes were palpable on the left side of the neck. The clinical diagnosis was a malignant tumor of the left lower gingiva. Positron emission tomography–computed tomography revealed uptake in the primary lesion (standardized uptake value [SUV] max: 18.89) and levels IB and IIA (SUVmax: 10.18). A biopsy was performed and a histopathological diagnosis of moderately to poorly differentiated SCC was made. The patient underwent left-sided segmental mandibulectomy, radical neck dissection, and fibula reconstruction. The histopathological diagnosis was invasive keratinizing SCC, pT4aN2bM0 (Grade 2, INFb, pHM0, pVM0, v1, ly1, neu2) (Figure 1a,b,d). No ENE was detected in level IB and IIA lymph nodes and postoperative CCRT was therefore performed. The patient’s prognosis was excellent, and he remained disease-free at the 75-month follow-up.

At 78 months after the first surgery, the patient revealed tongue ulceration on the right side, contralateral to his primary cancer. Careful investigation using the same methods as for his previous cancer confirmed advanced tongue cancer, as a second primary cancer. The previous CCRT meant that the patient had severe dysphagia, and he underwent total glossectomy, laryngectomy, modified radical neck dissection, and rectus abdominis musculo-cutaneous flap reconstruction. The histopathological diagnosis was invasive non-keratinizing SCC, pT4aN1M0 (Grade 2, INFc, pHM0, pVM0, v1b, ly1b, neu1C) (Figure 1a,c,e). There was no ENE in level IB lymph nodes. The histopathological findings of the second primary right-sided tongue cancer confirmed budding formation at the deepest point of tumor invasion compared with the primary left-sided mandibular gingival cancer. The patient remained cancer-free 48 months after surgery for the second primary cancer.

### 2.2. Establishment of Cultured Oral Metachronous Carcinoma Cell Lines HCM-SqCC020 (020) and HCM-SqCC020G (020G) from Each Tumor

The present study was conducted in accordance with the Declaration of Helsinki. The patient with OSCC provided consent in accordance with Hyogo Medical University institutional policies, and tumor samples were obtained according to a protocol approved by the Institutional Review Board of Hyogo Medical University (approval No. 267). Primary cell cultures of tumor cells and fibroblasts were obtained, as described previously [26,27,28].

Oral metachronous carcinoma cell lines arising from the same patient were established. The primary tumor cell line was termed HCM-SqCC020 (020) and the cell line from the second primary with TB expression was termed HCM-SqCC020G (020G) (Figure 2). The cell lines were passaged at least 50 times to confirm immortalization. The 020 and 020G cell lines were confirmed to derive from the patient’s tumor samples by short tandem repeat (STR) profiling using DNA extracted from each cell line and from the patient’s blood. Genotyping confirmed that the cell lines were of tumor origin and no contamination with other cell types was detected (Appendix A).

### 2.3. Exome Sequence Analysis

The primary gingival carcinoma and the second primary tongue carcinoma of this patient had different developmental stages and the tumor malignancy and invasion behaved completely differently. So, we investigated the tumor suppressor genes *TP53, CDKN2A*, and *NOTCH1*, which are involved in the regulation of cell differentiation, and *HRAS, PIK3CA, BRAF,* and *FAT1*, which are involved in tumor proliferation signals. They have previously been reported to have characteristically high mutation rates in OSCC [29,30,31]. We therefore focused on these genes and analyzed mutations that may affect their protein products in the 020 and 020G cell lines. *TP53*, a known cancer suppressor gene, was mutated in both 020 and 020G, but the position of the mutation on the chromosome differed (Table 1). *CDKN2A*, as a cancer suppressor gene, was only mutated in 020G. The *FAT1* gene, which is involved in the regulation of *Wnt* signaling related to cell proliferation and differentiation, was also mutated in 020G. We therefore analyzed the expression of the epithelial marker E-cadherin and yes-associated protein (YAP) as important factors involved in EMT and the Hippo pathway in 020 and 020G cells by Western blotting. YAP1 was upregulated and E-cadherin was downregulated in 020G cells with *FAT1* mutation compared with 020 cells (Figure 3). This suggests that the *FAT1* mutation affected the Hippo pathway and EMT induction in 020G cells.

### 2.4. Expression of p-EMT Marker at Deepest Site of Tumor Invasion and TB Expression

TB is a histopathologic finding that forms at the deepest site of tumor invasion. Cancer cells in the p-EMT state, as an intermediate state of EMT, acquire invasive potential and have both epithelial and mesenchymal properties, and are indistinguishable from cells expressing TB. Okuyama et al. reported that biomarkers of TB and p-EMT are common [32]. Therefore, assuming that the phenotype of p-EMT is TB, we hypothesized that the TB-positive cells in the deepest part of tumor invasion in the primary and second primary OSCC cells of this patient may also express the markers of p-EMT. We therefore performed immunohistochemical staining for TGF-β, which is thought to be a common marker for p-EMT and TB, and LAMC2, which is one of the markers for p-EMT.

TGF-β1 was expressed in the deepest tumor cells and was weakly positive in the stroma, and LAMC2 was also highly expressed in the deepest tumor cells. Interestingly, however, the late-stage tongue carcinoma with high expression of TB showed high expression of LAMC2 in TB-expressing cells and in tumor cells bordering the stroma, even in the tumor mass (Figure 4).

TGF-β1 is a marker of p-EMT and also an EMT inducer [18,33]. TGF-β1 expressed by tumor cells themselves or by the surrounding stroma may thus contribute to TB formation at the deepest site of tumor invasion.

### 2.5. Effects of TGF-β on Proliferative, Migratory, and Invasive Abilities, and Cell Morphology of 020 and 020G Cells

The TGF-β signaling pathway mediates and regulates the EMT process. TGF-β is a secreted protein that affects not only the cells that secrete the protein (autocrine) but also cells in its vicinity (paracrine), thereby inducing and maintaining EMT. TGF-β also contributes to cancer cell invasion by activating various matrix metalloproteinases (MMPs) involved in the degradation of the extracellular matrix [18,33,34,35]. We therefore hypothesized that when TGF-β, an EMT inducer, was added, EMT would be induced in 020 and 020G cells, causing cellular biological changes. To evaluate this, we assessed the effects of TGF-β on the proliferative, migratory, and invasive abilities of 020 and 020G cells, respectively, and observed changes in cell morphology.

TGF-β had no effect on the cell proliferative capacity of 020 or 020G cells (Figure 5), but the cells developed a spindle-shaped morphology after the addition of TGF-β (Figure 6). TGF-β also increased the migration ability of cells, especially 020G cells (Figure 7). These results suggest that TGF-β had no effect on the proliferative ability of either cell line, but induced EMT, which changed the cell morphology from cobblestone-like to a spindle shape, resulting in an increase in cell migration ability. TGF-β significantly increased the invasive ability of 020G cells, but not 020 cells (Figure 8). These results indicate that TGF-β induced EMT in both cell types, and increased the cell migration and invasion abilities of 020G cells more than 020 cells.

### 2.6. Evaluation of EMT and p-EMT in 020 and 020G Cells with and Without TGF-β

The above results suggested that TGF-β induced EMT in established 020 and 020G cells by causing a change in cell morphology to a spindle shape and increasing cell migration and invasion. We also hypothesize that the differences in the original tumor cell invasion patterns and TB expression between the two cell lines may be due to differences in the expression intensity of EMT-related genes when EMT is induced, resulting in differences in the degree of malignancy and invasion patterns. We therefore incubated 020 and 020G cells with TGF-β [18,33] for 48 h and then extracted and analyzed protein and mRNA levels of proteins involved in EMT, p-EMT, and EMT-TFs.

TGF-β decreased the expression of the epithelial marker E-cadherin and increased the mesenchymal marker N-cadherin, as shown by Western blot. There was no significant change in β-catenin, however, suggesting that this was not typical EMT. Expression levels of the p-EMT markers fibronectin and LAMC2 were also increased, and expression of phospho-SMAD2/3 was increased in response to SMAD2/3 downstream of the TGF-β pathway (Figure 9a).

TGF-β increased mRNA expression levels of the EMT-TFs *ZEB1/2*, *SNAIL2*, and *Twist2*, and the gelatinase group of MMPs, *MMP-9* and *MT1-MMP*, as shown by reverse transcription polymerase chain reaction (RT-PCR). Expression of the p-EMT marker *LAMC2* was also significantly increased. TGF-β, which acts as a transcriptional regulator, thus caused phosphorylation of SMAD2/3, induced the expression of EMT-TFs, regulated E-cadherin expression, and induced p-EMT, with increased expression of mesenchymal markers including N-cadherin, fibronectin, and LAMC2. Furthermore, cancer cells in the p-EMT state showed upregulated expression of MT1-MMP and MMP-9. These changes may result in basement membrane disruption (Figure 8) and contribute to increased invasive potential. Furthermore, p-EMT-related factors tended to be more highly expressed in 020G compared with 020 cells (Figure 9b).

## 3. Discussion

Despite improved functional outcomes as a result of advances in radiotherapy, chemotherapy, and surgical and imaging techniques, the survival rate for OSCC has only improved marginally over the past two decades [36], which is possibly associated with local recurrence and multiple oral cancers. Although there are no strict diagnostic criteria for oral metachronous carcinoma, it is generally accepted that the lesions must be located in different sites or must be contralateral in the same named site, or if ipsilateral, there must be no continuity between the two lesions and they must be clinically separated by at least 2.0 cm. Each lesion must also be histopathologically malignant. In terms of the interval of lesion development, McGuirt et al. [37] stated that treatment for the primary cancer must be completed and the secondary cancer discovered during the follow-up.

In the present case, the patient developed cancer of the right tongue on the opposite side more than 5 years after surgery and radiochemotherapy for left-sided mandibular gingival cancer. As noted above, the clinical and histopathological findings of the two tumors showed completely different findings, precluding a recurrence of gingival carcinoma, and the second cancer was therefore diagnosed as a metachronous carcinoma. Although metachronous multiple carcinomas have been diagnosed based on clinical findings, there have also been attempts to analyze their origin genetically, including genetic analysis of *p53* and other genes [38,39].

We therefore performed exome analysis of the 020 and 020G cell lines to confirm the genetic background of the metachronous carcinomas, focusing on mutations with a protein impact and an allele frequency ≤1% in Japanese individuals, among genomic abnormalities that are characteristic or frequently mutated in The Cancer Genome Atlas or in SCC of the head and neck. Referring to previous reports of *TP53* mutations in OSCC with hot spot missense mutations [40], only 020G cells had a hot spot mutation with c.245dupC; 020G cells also had mutations in *CDKN2A* and *FAT1*. The *CDKN2A* gene is located on chromosome 9p21 and contains an exonic region encoding the tumor suppressor protein p16, which prevents phosphorylation of the Rb protein and arrests cell cycle progression from the G1 to S phase. 020G cells had a *CDKN2A* mutation resulting in a stop codon, suggesting that the normal synthesis of p16 protein was not achieved and the cell cycle was not regulated, leading to secondary carcinogenesis. *CDKN2A* function has also been reported to be a prognostic marker for OSCC [41]. Changes in FAT1 have been shown to lead to defects in Hippo signaling and unrestricted YAP 1 activity, and upregulation of YAP is also known to induce EMT by causing downregulation of the epithelial marker E-cadherin and upregulation of the EMT-related TF *Snail2* [42,43,44,45]. *TP53* mutations were found in both 020 and 020G cells, but the details of the mutations were different: 020 had the c.173dupC; p.Gly59fs mutation, whereas 020G had the c.245dupC; p.Ala83fs mutation. In addition, mutations in the tumor suppressor genes *CDKN2A* (c.238C>T; p.Arg80* stop codon) and *FAT1* (c.9082-1G>C, c.8708C>T; p.Thr2903Ile) involved in the Hippo pathway [42] were only observed in 020G cells. The mutations in 020 and 020G cells were consistent with those found in the SCC of the head and neck. Finally, 020 and 020G were genetically proven to be de novo, not recurrent, tumors. *TP53*, *CDKN2A*, and *FAT1* are known tumor suppressor genes [29,30,31,46,47], and mutations in these genes were considered to be responsible for metachronous carcinogenesis through a lack of normal proteins. Loss of function of *FAT1* has also been reported to lead to activation of YAP [42]. Upregulation of YAP is also known to induce EMT by causing downregulation of the epithelial marker E-cadherin and upregulation of the EMT-related TF *Snail2* [44,45]. In the present study, YAP expression was significantly increased in 020G cells with the *FAT1* mutation, and RT-PCR showed higher expression of *Snail2* and significantly decreased expression of E-cadherin.

Despite progress in the diagnostic modalities and management of OSCC, the mortality rates remain dismally low, with a 5-year survival rate of <50%, thus questioning existing prognostic approaches [48,49]. Occult lymph node metastases are detected in up to 30% of patients with OSCC with a clinical N0 (cN0) neck on sentinel lymph node biopsy [50]. Identifying which early-stage OSCC patients are at higher risk of having occult lymph node metastases is critical for deciding whether to perform preventive neck dissection or preoperative therapy [6].

TB expression is also a significant risk factor for potential cervical lymph node metastasis in OSCC [16] and may be used as a factor for determining the need for cervical dissection, even in N0 cases without cervical lymph node metastasis. Further molecular biological analysis of TB expression is therefore needed to help determine the appropriate treatment for OSCC and to provide new prognostic factors and novel therapeutic targets.

RNA sequencing of OSCC identified TGF-β-induced p-EMT as a potential therapeutic target because TB cells express factors involved in TGF-β signaling and TB represents a transition to p-EMT phenotype [51]. TGF-β is an important promoter of p-EMT, regulating transcription of downstream target genes and activating downstream signaling pathways such as SMAD2/3 phosphorylation to induce EMT [52]. The current IHC results showed that TGF-β1 was expressed in tumor cells in the deepest area of tumor invasion, where TB was observed, and LAMC2 was expressed in TB-expressing cells and tumor cells bordering the stroma. In the 020 cell line established from a primary cancer (gingival carcinoma) that does not express TB, and in the 020G cell line established from a secondary cancer (tongue cancer) that does express TB, TGF-β expressed EMT-TFs and increased various p-EMT marker expression; however, typical EMT expression was not increased but p-EMTs were induced. TGF-β increased cell migration and invasive capacities, which are related to cell malignancy, in both cell types, but especially in 020G cells. 020G cells showed high expression of mesenchymal markers, and TGF-β increased the expression of p-EMT mesenchymal markers, but not typical EMT. Notably, expression levels of the EMT-TF *Snail2* and of *MT1-MMP* and *MMP-9* were increased. TGF-β treatment did not affect the proliferative capacities of the cells, but the migratory and invasive capacities were both higher in 020G cells. These findings suggest that TGF-β and other aspects of the deep TME may induce p-EMT in tumor cells, expressed as TB. Furthermore, TB cells have high migratory and invasive potentials and secrete gelatinases, such as MT1-MMP and MMP-2/9, suggesting that they are highly invasive in the stroma. LAMC2 as a marker of p-EMT is a laminin-5γ2 chain, and Peixoto da-Silva et al. [53] suggested that expression of a heterogeneous LN-5γ2 chain at the invasive front of tumors mediated the acquisition of a migratory and invasive epithelial cell phenotype in oral SCC. Moreover, Marangon et al. [54] reported that high-grade TB was associated with high expression of the cell–extracellular matrix adhesion molecule LN-5γ2 in tumor-peripheral stromal interactions in OSCC. The current Western blot and RT-PCR results also showed that TGF-β induced p-EMT and elevated LAMC2 expression in 020 and 020G2 cell lines.

This study has several limitations. Although we detected differences in expression levels of markers involved in p-EMT between cultured 020 (TB−) and 020G (TB+) cells derived from oral metachronous carcinomas in the same patient, this was an in vitro study. Further investigation of the markers involved in TB in vivo is thus required, including the development of an in vivo oral cancer xenograft model in immunodeficient mice by inducing p-EMT with TGF-β and using 3D cultures, which are more similar to the in vivo situation.

Based on the above, we successfully established cultured cell lines (020 and 020G) from oral metachronous carcinomas with different TB expression patterns from the same patient. These two cell lines were proven to be novel, rather than recurrent carcinomas, with different genetic mutations. Furthermore, the two cell lines showed different expression intensities of TGF-β-induced p-EMT, which may be useful for analyzing the relationship between p-EMT and TB.

## 4. Materials and Methods

### 4.1. Establishment of HCM-SqCC020 and HCM-SqCC020G Cell Lines

Briefly, tumor tissues were minced into 1–2 mm pieces using a disposable scalpel and placed on a culture dish in F-medium [55] with 10 μM Y-27632 (Wako Pure Chemicals, Osaka, Japan) for 1 week. The culture medium was then replaced with fresh medium, which was changed every 4 days thereafter. Tumor cells were separated from the primary mass culture using a magnetic-activated cell sorting (MACS^Ⓡ^ Miltenyi Biotic, Tokyo, Japan) system. The single-cell suspension of primary cultured cells was then centrifuged at 300× *g* for 10 min at room temperature and positive selection was performed using CD326 (EpCAM) MicroBeads^Ⓡ^ and an LS column Separator (Miltenyi Biotic, Tokyo, Japan), according to the manufacturers’ instructions. Once the cells reached confluence, they were washed with phosphate-buffered saline (PBS) (Mg^2+^- and Ca^2+^-free), detached with 0.05% ethylenediaminetetraacetic acid/trypsin for 5 min at 38 °C, and centrifuged (167× *g*, 4 °C, 5 min). Epithelial cells were resuspended in F-medium containing Y-27632 and plated at 0.3 × 10^6^ cells in a 60 mm dish. Two epithelial cell lines were successfully established from the patient sample, termed HCM-SqCC020 (020: first primary) and HCM-SqCC020G (020G: second primary), respectively. Both cell lines were incubated in F-medium [52] with 10% fetal bovine serum (FBS) at 37 °C and 5% CO_2_. The morphologies of the exponentially proliferating cells in monolayers were reviewed and documented by inverted phase contrast microscopy.

### 4.2. Short Tandem Repeat Authentication of HCM-SqCC020 and SqCC020G Cell Lines

The identities of the cell lines were verified by extracting genomic DNA from the blood of the patient whose tumor sample was used to generate the cell lines, and from the cells themselves, using a QIAamp DNA Mini Kit (Qiagen, Venlo, The Netherlands), according to the manufacturer’s protocol. The DNA underwent genotyping by STR profiling by BEX Co., Ltd. (Tokyo, Japan), using a GenePrint 10 System (Promega, Madison, WI, USA) and a 3130xl Analyzer (Applied Biosystems, Foster City, CA, USA).

### 4.3. Exome Capture and Sequencing

Genomic DNA obtained from each sample, as described above, was subjected to exome capture using an Agilent SureSelect Human All Exon Kit V6 (Agilent Technologies, Inc., Santa Clara, CA, USA) (target size: 60.5 megabases), according to the manufacturer’s protocols. The equally pooled libraries of the samples were sequenced using the NovaSeq X Plus system (Illumina, San Diego, CA, USA) in 151-base-pair (bp) paired-end reads. All of the analyses were carried out by DNA Chip Research Inc. (Kanagawa, Japan).

### 4.4. Exome Sequencing

Alignment and Somatic Variant Calling

The sequence reads were aligned to the human reference genome (GRCh38/hg38 + decoy sequences [hs38d1]) obtained from the Genome Analysis Toolkit (GATK) resource bundle (https://gatk.broadinstitute.org/hc/en-us/articles/360035890811-Resource-bundle accessed on 19 July 2024) using the Burrows–Wheeler Aligner, version 0.7.17 [56]. Multiple identical reads from the same fragment were marked as duplicates and removed using GATK version 4.2.0 (https://gatk.broadinstitute.org/hc/en-us accessed on 19 July 2024). Somatic variant calling in tumor/normal sample pairs was performed with Strelka somatic variant caller, version 2.9.10 [57], and Manta structural variant and indel caller, version 1.6.0 [58], with default settings as recommended by the best-practice workflow (https://github.com/Illumina/strelka/tree/v2.9.x/docs/userGuide accessed on 19 July 2024). All these analyses were carried out by DNA Chip Research Inc. (Kanagawa, Japan).

2.Variant Annotation and Filtering

Functional annotations of the Ensembl database GRCh38.99 [59] and the possible effects of variants were added using SnpEff version 5.0E [60]. Based on these annotations, the variants were first filtered for those predicted to alter amino acid sequences (missense, nonsense, and splice-site mutations and indels in coding regions), and then for rare variants (<1.0% minor allele frequency in gnomAD version 2.1.1 JPN [76 Japanese individuals] or the Human Genetic Variation Database [61], which contained genetic variations determined by whole-exome sequencing in 1208 Japanese individuals). All these analyses were carried out by DNA Chip Research Inc. (Kanagawa, Japan).

We analyzed mutations in the 020 and 020G cell lines resulting in proteins with potential impacts on *TP53*, *CDKN2A*, *NOTCH1*, *HRAS*, *PIK3CA*, *BRAF*, and *FAT1*, which are known to have high frequencies of mutations in OSCC [29,30,31,46].

### 4.5. IHC to Evaluate TB

Formalin-fixed, paraffin-embedded OSCC specimens were sectioned at 4 µm thickness and stained by IHC to investigate the association between p-EMT and TB. The following primary antibodies were used for IHC: anti-pan cytokeratin antibody (ab86734; dilution 1:100; Abcam, Inc., Cambridge, UK), anti-TGF-β1 antibody (Y241; dilution 1:100; Yanaihara Laboratories, Fujinomiya, Japan), and anti-LAMC2 antibody (ab210959; dilution 1:500; Abcam, Inc.). All primary antibodies were incubated overnight with the tissue at the specified concentrations. TB was defined as a single cell or a cell cluster of up to five tumor cells at the invasive tumor front, according to Hong et al. [19]. Tissues were then incubated with Dako EnVision™^+^ Dual Link System-HRP (Dako, Tokyo, Japan) secondary antibody for 30 min at room temperature. Protein expression was visualized using a 3,3′-diaminobenzidine tablet (Dako) [19]. TB was selected for evaluation in each case, and the number of buds was counted at ×200 magnification.

### 4.6. Cell Proliferation Assay

The 020 and 020G cells (5000 cells/well) were plated in 96-well plates with or without 10 ng/mL TGF-β and cell counts were analyzed after 6, 18, 30, and 42 h using a Cell Counting Kit-8 (Dojindo Molecular Technologies, Kumamoto, Japan). After incubation with the reagent at 37 °C, the optical density was read at 450 nm using a Benchmark Plus^Ⓡ^ microplate reader (BIO-RAD, Hercules, CA, USA).

### 4.7. Scratch Assay

Confluent monolayers were prepared by plating the 020 and 020G cells in 60 mm dishes. The cell monolayers were then scraped off in a straight line using a pipette tip. The cells were washed once followed by the addition of 10 ng/mL TGF-β and observed in the same field of view by a phase contrast microscopy. The cells were incubated at 37 °C for 24 h and the distance of each scratch closure was measured [62].

### 4.8. Invasion Assay

Invasion assays were performed using a CytoSelect™ 24-well cell invasion assay kit, according to the manufacturer’s instructions (Cell Biolabs, San Diego, CA, USA) [63] (Figure 10). The 020 and 020G cell suspensions (1.0 × 10^6^ cells/) were prepared in serum-free medium and 500 μL of medium containing 10% FBS and 10 ng/mL TGF-β was then added to the lower well of the invasion plate, partially to examine the effect of TGF-β on cell invasion ability. Cell suspension (300 µL) was then added to the inside of each insert and the plates were incubated for 24 h at 37 °C in 5% CO_2_. The culture medium was then aspirated carefully from the inside of the insert; the ends of two or three cotton-tipped swabs were soaked with water and flattened by pressing against a clean hard surface and the interiors of the inserts were swabbed gently to remove non-invasive cells. Each insert was then transferred to a clean well containing 400 µL Giemsa and incubated at room temperature for 10 min. The stained inserts were washed gently several times in a beaker of water and allowed to air dry. Invasive cells were counted under a light microscope at high magnification (×100), with three individual fields per insert (Figure 10).

### 4.9. Western Blotting

After culturing, the cells were washed with PBS (Mg^2+^- and Ca^2+^-free) and centrifuged. RIPA buffer (cat. No. sc-24948; Santa Cruz, Inc., Dallas, TX, USA) was added and the cells were incubated at 4 °C for 60 min and then centrifuged again at 12,000× *g* for 20 min at 4 °C. The supernatant was used as the total cell lysate. Proteins were extracted from the lysate and the concentrations were measured using the Bradford assay [64]. Western blotting was performed as described previously [65] and signals were detected by chemiluminescence, using a Pierce SuperSignal Western blotting kit (Thermo Fisher Scientific, Waltham, MA, USA). The primary and secondary antibodies are shown in Appendix A.

### 4.10. RNA Extraction and RT-PCR Analysis

RT-PCR was performed as described previously [28,66]. Briefly, RNA was extracted using TRIzol (Invitrogen, Carlsbad, CA, USA) and reverse transcribed using a Prime SCRIPT RT-PCR kit (Takara Bio, Kusatsu, Japan), according to the manufacturer’s instructions. The reagents were adjusted, denatured, and annealed at 65 °C for 5 min and then further adjusted; next, the mixture was denatured and annealed at 30 °C for 10 min, 42 °C for 30 min, and overnight at 4 °C. The relevant primers were prepared the next day, and the reaction was repeated for 35 cycles of 10 s at 98 °C, 10 s at 60 °C, and 1 min at 68 °C. The sequences of the primers are shown in Appendix A. *GAPDH* was amplified as a control.

### 4.11. Statistical Analysis

For all data sets, significant differences were evaluated by Mann–Whitney U tests, with a *p* < 0.05 considered statistically significant.

## 5. Conclusions

We successfully established cultured cell lines from metachronous oral carcinomas with differential TB expression arising from the same patient. These two cell lines differ in terms of their molecular biology, migration, invasive ability, and expression of markers related to p-EMT, and thus have the potential to be utilized to elucidate the relationship between TB and p-EMT.

## Figures and Tables

**Figure 1 ijms-26-03347-f001:**
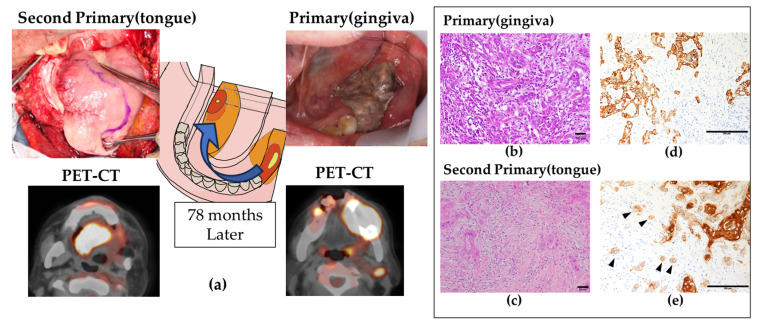
Overview and histopathological findings of oral metachronous carcinoma arising in the same patient. (**a**) Intraoral findings of metachronous oral carcinoma and positron emission tomography–computed tomography images. (**b**,**c**) Hematoxylin–eosin-stained histopathological findings of gingival carcinoma, the primary cancer (**b**), and tongue carcinoma, the second primary cancer (**c**). (**d**,**e**) Immunohistochemical staining of AE1/AE3 at the deepest invasive site of gingival carcinoma (**d**) and tongue carcinoma (**e**). Arrows indicate TB cells. (**b**–**e**): magnification: ×200.

**Figure 2 ijms-26-03347-f002:**
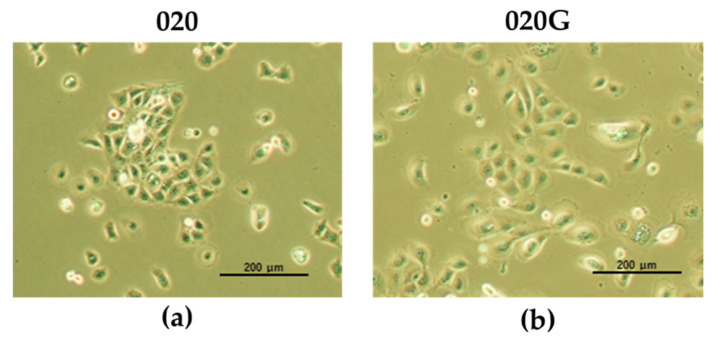
Established oral metachronous carcinoma cell lines. (**a**) HCM-SqCC020 cell line established from primary gingival carcinoma (TB−). (**b**) HCM-SqCC020G cell line established from second primary tongue cancer (TB+). Magnification: ×100.

**Figure 3 ijms-26-03347-f003:**
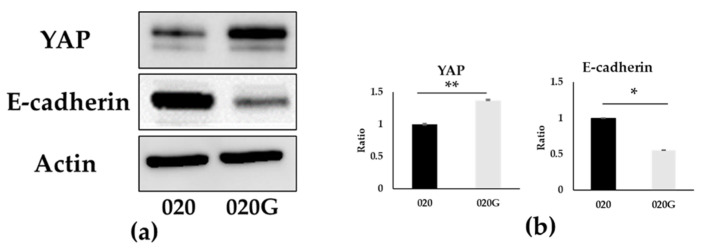
(**a**) Western blotting analysis of YAP, E-cadherin, and actin in 020 and 020G cells; (**b**) analysis using densitometry. 020G cells with *FAT1* mutation showed higher YAP expression and reduced E-cadherin expression compared with 020 cells (* *p* < 0.05, ** *p* < 0.01).

**Figure 4 ijms-26-03347-f004:**
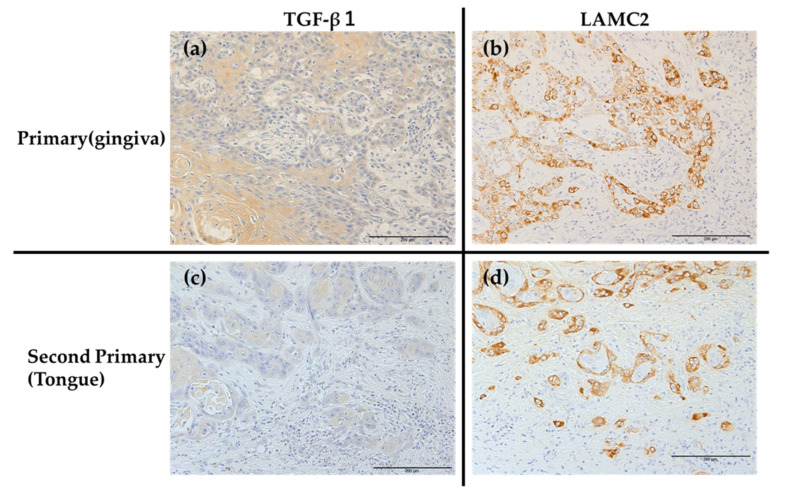
(**a**–**d**) Immunohistochemical staining of TGF-β1 and LAMC2 at the deepest site of tumor invasion in primary gingival carcinoma and second primary tongue cancer. Scale bar: 200 µm.

**Figure 5 ijms-26-03347-f005:**
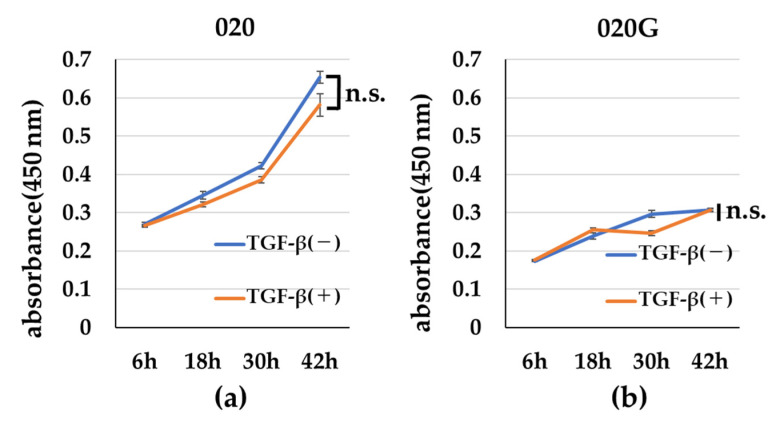
Cell proliferative capacity with or without TGF-β. Incubation of (**a**) 020 and (**b**) 020G cells with TGF-β for 42 h had no effect on their proliferative capacity. Bars indicate standard deviation. n.s., not significant. *p* ≥ 0.05 compared with cells cultured without TGF-β.

**Figure 6 ijms-26-03347-f006:**
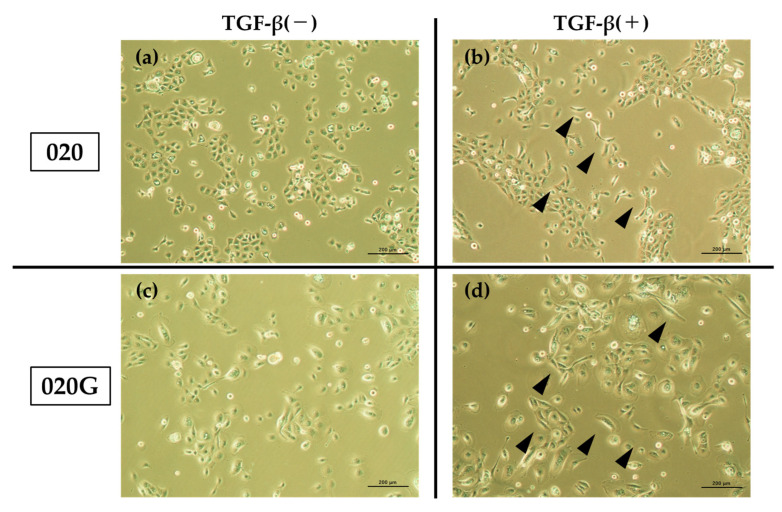
Morphological changes in cells induced by TGF-β. Both cells developed a spindle-cell morphology after the addition of TGF-β. (**a**,**b**) 020 cells: (**a**) TGF-β(−); (**b**) TGF-β(+). (**c**,**d**) 020G cells: (**c**) TGF-β(−); (**d**) TGF-β(+). Arrow head: Cells that have changed into spindle-shaped cells. Magnification ×100.

**Figure 7 ijms-26-03347-f007:**
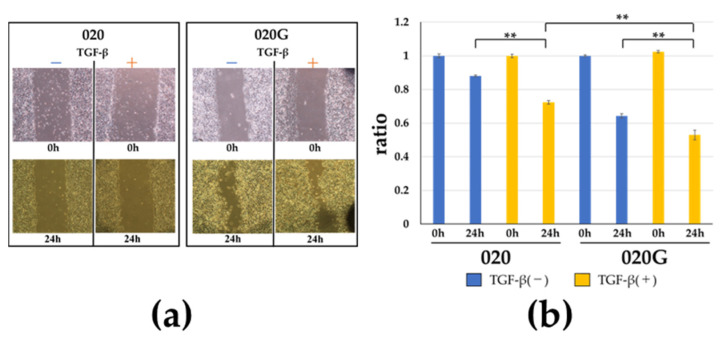
(**a**) Effect of TGF-β on cell migration of 020 and 020G cells. (**b**) Cell migration abilities of 020 and 020G cells with and without TGF-β. Percentage reduction in width of scratched area relative to time of scratching. TGF-β increased cell migration ability, and the effect was significant in 020G cells. ** *p* < 0.01.

**Figure 8 ijms-26-03347-f008:**
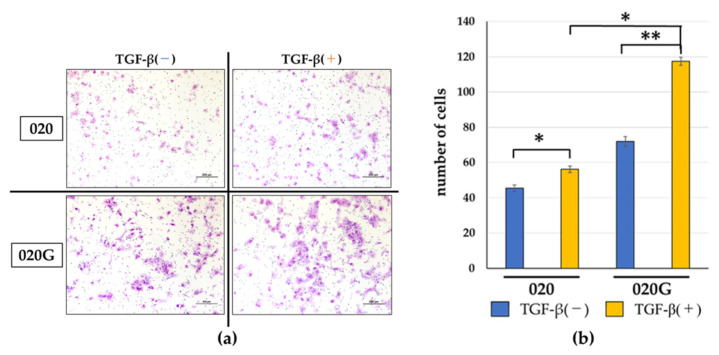
(**a**) Cell invasion capacities of 020 and 020G cells with or without TGF-β. TGF-β increased cell invasion in both cell types, but the effect was greater in 020G cells. Scale bar, 200 μm. (**b**) Numbers of cells invading the membrane. TGF-β increased the invasive capacities of both cell lines, but the effect was more significant for 020G cells. * *p* < 0.05, ** *p* < 0.01.

**Figure 9 ijms-26-03347-f009:**
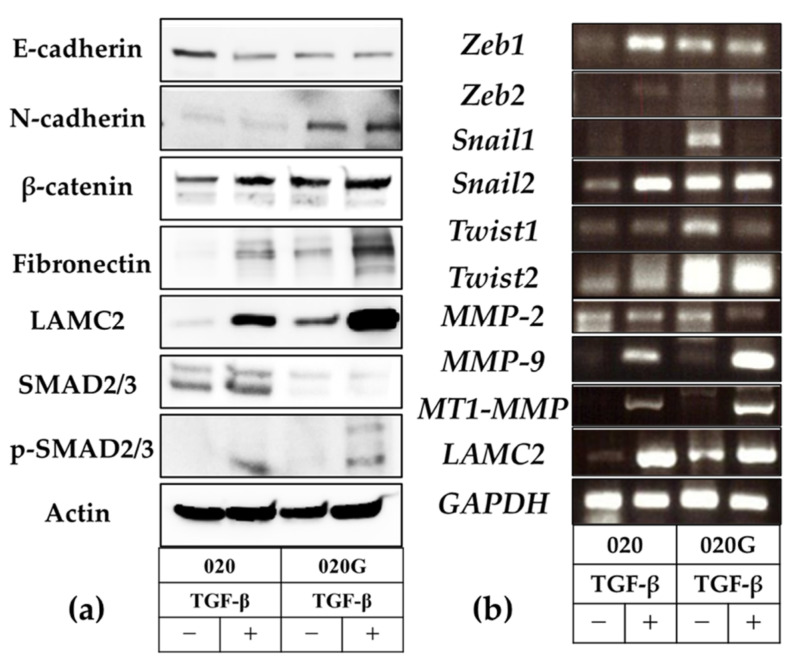
(**a**) Analysis of epithelial–mesenchymal marker expression by Western blotting and evaluation of p-EMT. TGF-β decreased expression of the epithelial marker E-cadherin and increased expression of the mesenchymal markers N-cadherin, β-catenin, fibronectin, and LAMC2, as markers of p-EMT. SMAD2/3, a signaling molecule downstream of TGF-β, was phosphorylated and p-SMAD2/3 expression was increased, indicating EMT induction. (**b**) TGF-β enhanced the expression of the gelatinase group of MMPs or the EMT-TF system, as shown by RT-PCR. *LAMC2* expression was also enhanced.

**Figure 10 ijms-26-03347-f010:**
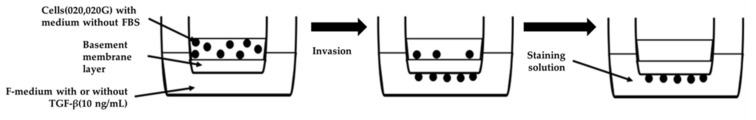
The cell suspension was placed in the upper chamber and infiltrating cells passed through the basement membrane layer and adhered to the bottom of the insert membrane. After removing non-infiltrating cells, infiltrating cells were stained with Giemsa stain and quantified.

**Table 1 ijms-26-03347-t001:** Exome analysis in 020 and 020G cells (* Stop codon).

**Cell**	**CHROM**	**POS**	**REF**	**Variant Type**	**Allele**	**Annotation**	**Gene Name**	**HGVS.c**	**HGVS.p**
020	chr17	7676195	T	INS	TG	Frameshift variant	*TP53*	c.173dupC	p.Gly59fs
020G	chr4	186614345	C	SNV	G	Splice acceptor variant and intron variant	*FAT1*	c.9082-1G>C	
	chr4	186617884	G	SNV	A	Missense variant	*FAT1*	c.8708C>T	p.Thr2903Ile
	chr9	21971121	G	SNV	A	Stop gained	*CDKN2A*	c.238C>T	p.Arg80*
	chr17	7676123	C	INS	CG	Frameshift variant	*TP53*	c.245dupC	p.Ala83fs

## Data Availability

Data are contained within the article.

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
