# Peer review of "Mechanism of Tumor Budding in Patient-Derived Metachronous Oral Primary Squamous Cell Carcinoma Cell Lines"

_ijms, 2025, doi:10.3390/ijms26073347_

Round 1

Reviewer 1 Report

Comments and Suggestions for Authors

The manuscript presents an investigation into the role of tumor budding (TB) and partial epithelial-mesenchymal transition (p-EMT) in OSCC using patient-derived cell lines. The study offers valuable insights into the molecular mechanisms underpinning TB and its potential as a prognostic biomarker in OSCC. 

Here are some points to improve the clarity of the manuscript.

Introduction

1- The introduction lacks a clear explanation of why TB and p-EMT were specifically selected for study in OSCC. Please discuss TB in light of the discussion provided by this paper. Histol Histopathol. 2024 Jan;39(1):1-12. doi: 10.14670/HH-18-634. 

2- The provided list of genes involved in EMT and TME need further refinement and articulation with the latest understanding of the molecular landscape of oral squamous cell carcinoma. The authors are recommended to discuss key previous work on TB and EMT in OSCC, which is unclearly described. 

3- The manuscript would benefit from a more precise hypothesis-driven approach rather than a descriptive exploration.

4- The authors didn't explain their selection of a limited set of genes (TP53, CDKN2A, NOTCH1, HRAS, PIK3CA, BRAF, FAT1) rather than performing a global unbiased analysis. The authors are advised to conduct and report a pathway enrichment analysis (e.g., KEGG, GO) to determine whether additional pathways are altered in TB-positive vs. TB-negative cells.

5- TB is a histopathological phenomenon observed in tissue sections, and its recreation in 2D culture is questionable. Can the authors explain the rationale for 2D experiments rather than conducting experiments in animal or 3D models that capture tumour microenvironment. 

6- Western blot images are of low resolution and need better contrast. Please provide densitometric quantification of Western blots normalized to housekeeping proteins.

7- Please add scale bars and proper annotations to microscopy images.

8- Discussion needs some refinement in which the authors should avoid overstretching their findings since there no functional assasy, specifically about YAP role. 

Comments on the Quality of English Language

The English language is of acceptable standard. 

Author Response

We are grateful to you for your time, effort, thoughtful suggestions.

We confirm that this manuscript has been improved by correcting the points you have indicated.

Indicating point 1

The introduction lacks a clear explanation of why TB and p-EMT were specifically selected for study in OSCC. Please discuss TB in light of the discussion provided by this paper. Histol Histopathol. 2024 Jan;39(1):1-12. doi: 10.14670/HH-18-634.

Response: Thank you for your opinion. At the beginning of this article, I mentioned that TB was recently found to be a predictor of cervical lymph node metastasis, which is a prognostic factor for OSCC, based on previous studies. This means that the elucidation of TB is important for the therapeutic strategy of OSCC. In a previous study, Puram et al. reported that p-EMT is not a classical EMT program, but rather a plastic metastable state that has both mesenchymal properties while retaining epithelial markers of adhesion. They also reported that the p-EMT program is localized at the tip of the primary tumor, enabling collective migration of tumor cells, and is in close proximity to tumor-associated fibroblasts in the TME. Based on this, we thought that tumor cells express TB as a phenotype of p-EMT, and we came up with the idea of inducing p-EMT in cultured cell lines established from oral metachronous carcinomas that developed in the same human individual, which have differences in TB expression, and evaluating the intensity of p-EMT expression and the degree of malignancy of the tumors. We have added this contents to lines 69 to 80 of the introduction.

Indicating point 2

The provided list of genes involved in EMT and TME need further refinement and articulation with the latest understanding of the molecular landscape of oral squamous cell carcinoma. The authors are recommended to discuss key previous work on TB and EMT in OSCC, which is unclearly described.

Response: Thank you for your opinion. The cancer microenvironment (TME) includes various elements such as cancer-associated fibroblasts, tumor-associated macrophages, vascular and lymphatic cells, and hypoxia. Liquid factors such as TGF-β from cells in the TME and hypoxic conditions around tumors are thought to play tumor-promoting roles such as induction of EMT. Recently, results of single-cell RNA-Seq of HNSCC for analysis of the EMT expression program have been shown, including the classical EMT markers podoporanin (PDPN), vimentin (VIM), integrin alpha, and extracellular matrix genes such as MMPs, integrins and laminin, TGF-β were included (Reference 22). EMT-like cells furthermore maintain expression of multiple cytokeratins and other epithelial markers and initially undergo induced changes in EMT via upregulation of Snail2, suggesting that this may be p-EMT in a hybrid state. Therefore, TGF-β has been found to be a potent agent in the dynamic transition of cells to p-EMT. TB cells, on the other hand, have cytoplastic properties such as cytoskeletal deformability and motility, which have been shown to be associated with decreased expression, but not complete loss of E-cadherin, and increased expression of mesenchymal markers (Reference 32), suggesting a close relationship between TB and p-EMT. In our study, we found high expression of LAMC2, one of the p-EMT markers, in TB-positive cells and in the TB-positive cells, and increased expression in WB and RT-PCR, suggesting a link between TB and p-EMT. Other p-EMT markers, such as LAMB3, were also examined, but LAMC2 was the most prominent.

.

Indicating point 3

The manuscript would benefit from a more precise hypothesis-driven approach rather than a descriptive exploration.

Response: Thank you for your comment. The background and hypotheses that led to this research are also described in the responses to points 1 and 2.

In addition, the following has been inserted to add to the hypotheses for each research content.

Line 168~172, Line 190~194, Line 227~230.

Indicating point 4

The authors didn't explain their selection of a limited set of genes (TP53, CDKN2A, NOTCH1, HRAS, PIK3CA, BRAF, FAT1) rather than performing a global unbiased analysis. The authors are advised to conduct and report a pathway enrichment analysis (e.g., KEGG, GO) to determine whether additional pathways are altered in TB-positive vs. TB-negative cells.

Response: Thank you for your opinion. The aim of the exome analysis conducted in this study was to analyze the exon regions of the DNA extracted from the blood samples of patients with 020 and 020G, to clarify their genetic backgrounds, such as whether there are common or unique gene mutations in 020 and 020G, and to clearly distinguish between recurrent cases and de novo cases. However, few genetic mutations common to 020 and 020G were found. Given the fact that the first gingival carcinoma and the later tongue carcinoma of this patient had different developmental stages, and that the tumor malignancy and invasion behaved completely differently, as shown in Reference 29-31, we investigated the tumor suppressor genes TP53 and CDKN2A to confirm whether the tumors were recurrence or de novo, NOTCH1, which is involved in the regulation of cell differentiation, and HRAS, PIK3CA, BRAF, FAT1, which is involved in tumor proliferation signals (added to 140 to 146 in the text). It was confirmed that 020 and 020G had different TP53 mutation sites, and only 020G also has CDKN2A and FAT1 gene mutations, confirming that it was a heterochronic cancer from the same human individual. We confirmed that this is a cultured oral metachronous carcinoma cell line derived from the same human individual.

Indicating point 5

TB is a histopathological phenomenon observed in tissue sections, and its recreation in 2D culture is questionable. Can the authors explain the rationale for 2D experiments rather than conducting experiments in animal or 3D models that capture tumour microenvironment.

Response: Thank you for your opinion. In this study, we established 020 and 020G cell lines from gingival cancer (TB-) and tongue cancer (TB+), respectively. The difference in TB expression between 020 and 020G is thought to be due to the difference in the intensity of p-EMT. As explained in the results of this study, stromal cells induce p-EMT and TB expression in tumor cells, but it is challenging to develop a histopathological TB model because stroma makes a complex approach to tumor cells, so we attempted to reproduce it using a conventional EMT induction model in a 2D environment. We added TGF-β, a typical p-EMT inducer, and examined whether there were any changes in the behavior, morphology, migration, invasion, and proliferation of 020 and 020G cells when p-EMT was induced. When TGF-β was added to 2D culture, the 020G cultured cell line actually showed increased expression of mesenchymal markers and increased invasion and migration. In other words, in this study, we established a culture cell line that is useful for TB analysis using the same human-derived culture oral heterotopic carcinoma cell line.

Indicating point 6

Western blot images are of low resolution and need better contrast. Please provide densitometric quantification of Western blots normalized to housekeeping proteins.

Response: Thank you for your suggestion. Thank you for your comment. As you pointed out, we processed the WB images with high contrast. We quantitatively evaluated the changes in the density of each of the 020 and 020G with TGF-β added, using the TGF-β (-) data as a standard. The results are attached to the Figure Legends, and we have also added a statistical analysis. We would be grateful if you could check the Figure Legends.

Indicating point 7

Please add scale bars and proper annotations to microscopy images.

Response: Thank you for your suggestion. We have added a scale bar to the microscope image.

Indicating point 8

Discussion needs some refinement in which the authors should avoid overstretching their findings since there no functional assasy, specifically about YAP role.

Response: Thank you for your comments. As you pointed out, we did not perform a functional assay of YAP in this study, so we have revised the main text so that it does not lead to an over-broad interpretation of the data (Lines 292 to 297 in the text).

Reviewer 2 Report

Comments and Suggestions for Authors

This study reports the establishment and characterization of two primary tumor cell lines from two metachronous oral squamous cell carcinomas (OSCC). Specifically, primary tumor cell lines were obtained from a gingival OSCC (020 cells) and a tongue OSCC (020G cells). The tongue OSCC arose in the same patient more than 6 years after the gingival OSCC. Of note, compared to OSCC gingivalis, OSCC of the tongue had higher levels of tumor budding. Genetic analyses revealed that the tumor suppressor gene TP53 was mutated in both 020 and 020G cells, although the location of the mutation on the chromosome differed. Unlike TP53, the CDKN2A and FAT 1 genes were mutated in 020G cells, but not in 020 cells. The results of protein analyses indicated that YAP1 was upregulated and E-cadherin was downregulated in 020G cells compared with 020 cells. Finally, functional assays showed that 020G cells were more responsive to the pro-EMT and pro-invasive effects of TGF-beta 1 than 020 cells.  

This study is original and relevant for the field, and adds useful information to understand the mechanisms for OSCC budding. The methodology adopted is correct and all appropriate controls have been included. The conclusions are consistent with the evidence and arguments presented, the references are appropriate and the quality of the data and illustrations is good.

Author Response

We also appreciate the time and effort you and each of the reviewers have dedicated to providing insightful feedback on ways to strengthen our paper. Thus, it is with great pleasure that we resubmit our article for further consideration. We also hope that our edits and the responses we provide satisfactorily address all the issues and concerns you and the reviewers have noted.